

# Atmospheric data set from the Geodetic Observatory Wettzell during the CONT-17 VLBI campaign

Thomas Klügel[1], Armin Böer[1], Torben Schüler[1], Walter Schwarz[1]

5    [1]Federal Agency for Cartography and Geodesy, Geodetic Observatory Wettzell, 93444 Bad Kötzting, Germany

*Correspondence to*: Thomas Klügel (thomas.kluegel@bkg.bund.de)



**Abstract**

Continuous Very Long Baseline Interferometry (VLBI) observations are designed to obtain highly accurate data for detailed studies of high frequency Earth rotation variations, reference frame stability and daily to sub-daily site motions. During the CONT-17 campaign that covered a time span of 15 days between Nov. 28 and Dec. 12, 2017, a comprehensive data set of

atmospheric observations and weather model data were acquired at the Geodetic Observatory Wettzell, where three radio telescopes contributed to three different networks which have been established for this campaign. The data set is made available to all the interested users in order to provide an optimal data base for the analysis and interpretation of the CONT-17 VLBI data. In addition, it is an outstanding data set for validation and comparison of tropospheric parameters resulting from different space techniques with regard to the establishment of a common atmosphere at co-location sites.

The regularly recorded atmospheric parameters comprise all meteorological quantities (pressure, temperature, humidity, wind, radiation, precipitation) taken from the local meteo station close to the surface, solar radiation intensity, temperatures up to 1000 m above the surface from a temperature profiler, total vapour and liquid water content from a water vapour radiometer, and cloud coverage and -temperatures from a nubiscope. Additionally, vertical profiles of pressure, temperature and humidity from radiosonde balloons and from numerical weather models were used for comparison and validation.

The graphical representation and comparison show a good correlation in general, but also some disagreements at special weather situations. While the accuracy, the temporal and spatial resolution of the individual data sets is very different, the data as a whole characterize comprehensively the atmospheric conditions around Wettzell during the CONT-17 campaign and represent a sound basis for further investigations (https://doi.pangaea.de/10.1594/PANGAEA.895518)

# 1 Introduction

## 1.1 Geodetic VLBI observations and CONT continuous measurement campaigns

The International VLBI (Very Long Baseline Interferometry) Service for Geodesy and Astrometry (IVS) is coordinating geodetic VLBI observing programmes (Schlüter and Behrend, 2007). VLBI is important since it is the only geodetic technique capable to derive the full set of Earth orientation parameters (EOP). The IVS organises special measurement campaigns called 'CONT' approximately each 3 years since 2002. These particularly intensive sessions cover two weeks of

continuous network observations and must be distinguished from the routine observation programme consisting of individual 24 h and 1 h sessions. The main goal of CONT is to probe the accuracy of the VLBI estimates of the Earth orientation parameters and to investigate possible network biases (Behrend et al., 2017). The CONT17 campaign started on November, 28[th] 2017 with observations being carried out in three different networks (Behrend, 2017).

## 1.2 Geodetic Observatory Wettzell and purpose of atmospheric observations

The Geodetic Observatory Wettzell (GOW) features two SLR (Satellite Laser Ranging) telescopes, several GNSS (Global Navigation Satellite System) reference stations, a DORIS (Doppler Orbitography and Ranging Integrated by Satellite)



beacon as well as three VLBI telescopes (Schüler et al., 2015). All three radiotelescopes participated in CONT17, each of them in one of the three different networks. VLBI, GNSS as well as DORIS all operate in the microwave frequency domain. In this case, the atmosphere is a major complicating factor reducing the accuracy (Petit and Luzum, 2010). Consequently, the set of atmosphere sensors at the Geodetic Observatory Wettzell was substantially enhanced in recent years to provide means

to better deal with this problem. The propagation delays induced by the ionosphere can be compensated with help of measurements taken on at least two different frequencies.

However, the troposphere (and to a lesser extent also the stratosphere) remains a problem. The microwave signals are delayed when passing through these layers, and these effects are non-dispersive, i.e. virtually identical on the various frequencies in use. As a consequence, pre-elimination of these propagation errors is not possible. One method to quantify

troposheric errors is to use models. Another one is to introduce tropospheric unknowns as nuisance parameters into the observation equations, and to estimate these effects together with the set of target parameters. In practice, a combination of both approaches is usually accomplished. In any case, real measurements of the state of the atmosphere are very valuable to aid in tropospheric delay modelling and to interpret the results and residuals. This is the motivation to compile the atmosphere measurements collected during the CONT17 campaign forming a comprehensive data set to understand the

atmosphere over the Geodetic Observatory Wettzell and to aid VLBI analysis.

## 2 Study area and instrumentation

The Geodetic Observatory Wettzell is located in Eastern Bavaria on a flat montain ridge about 600 m above sea level, that is, above standard elevation zero (NHN) of the German height system (DHHN). The topography in the surroundings ranges from valley floors (~400 m a.s.l.) to mountain ridges (~1000 m a.s.l.). Land coverage is characterized by grassland and

forest. A plan view of the observatory with the instrument locations is depicted in Fig. 1. The following sections give a description of the deployed instruments and the measured quantities.



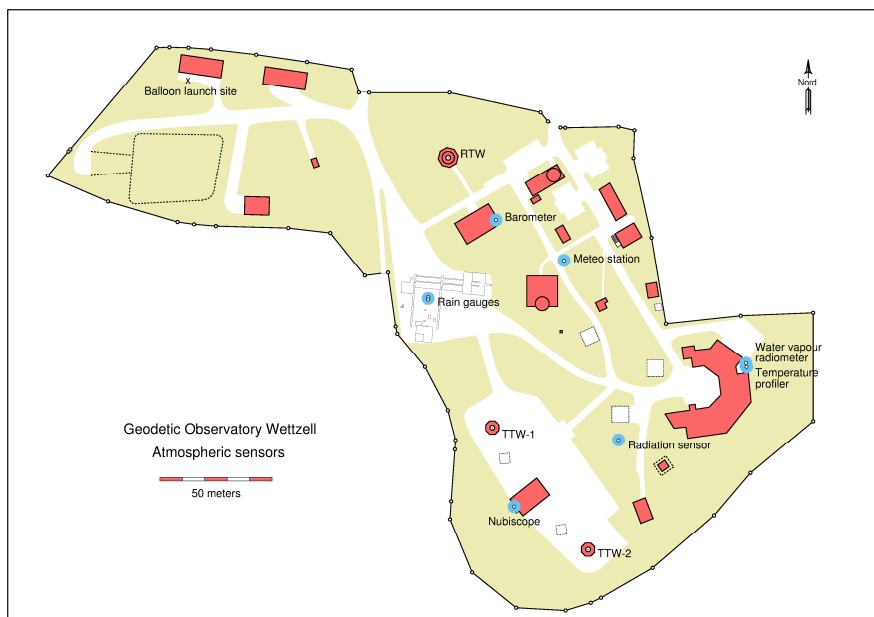

Figure 1: Geodetic Observatory Wettzell with atmospheric sensors highlighted in blue.

### 2.1 Local weather station

The temperature, humidity and wind sensors of the local weather station are mounted on a concrete tower at 7 m and 10 m

5  height above the surface (Table 1). The air pressure sensor is inside the RTW control building and the rain gauges are mounted on a platform as shown in Fig. 1. Data are continuously acquired and averages are recorded once per minute. For wind direction and wind speed, minimum and maximum values being measured within 1 minute are also stored, indicated by "<" and ">". The heated rain gauges measure snow as well and record the sum over 1 minute.

10  **Table 1. Sensors of the local weather station. Except for the pressure sensor, the height is given in meters above the surface.**

| Sensor: | WD | WS | T1 | T2 | RH1 | RH2 | SM | P | R1 | R2 |
|---|---|---|---|---|---|---|---|---|---|---|
| Height: | 10 m | 10 m | 10 m | 7 m | 10 m | 7 m | -0,5 m | 609.3 m a.s.l. | 1 m | 1 m |
| Mesured quantity: | Wind direction | Wind speed | Air temperature | | Air Humidity | | Soil moisture | Air pressure | Precipitation | |
| Accuracy: | 0.2 ° | 0.1 m/s | 0.1 °C | | 2.5 % | | 2 % | 0.01 hPa | 10 % | |
| Type: | Lambrecht 14512 G3 | | Lambrecht 809 MU | | Lambrecht 809 MU | | TRIME-EZ | Paroscientific 740–16B | L-Tec | |



## 2.2 Radiation sensor

As an addition to the meteo station, the global radiation is measured using a pyranometer Thies CM 11. At the same place a net radiometer (Kipp & Zonen NR Lite) measures the difference between radiation from above, i.e. the sun and the sky, and

from below, i.e. the soil surface. Both sensors are installed 1.5 m above the grass surface. The sampling rate is 10 minutes.

## 2.3 Temperature profiler

A quasi-continuous record of temperatures in the atmosphere up to 1000 m height is realized by a radiowave radiometer MTP-5 from R.P.O. Attex. The microwave receiver measures the blackbody thermal radiation of the atmosphere at a frequency of 56.6 GHz. The intensity of the radiation is a function of the temperature. By scanning the atmosphere at

different elevation angles, the operating software computes temperatures at different heights in 50 m steps up to 1000 m under the assumption of a horizontal temperature layering. The basic principle and some field examples are described in Peña et al. (2013).

The temperature profiler is installed on a tower at 619 m a.s.l. and 10 m above ground. A complete profile is recorded each 5 minutes. The accuracy is specified with 0.2 to 1.2 °C, depending on the profile type and height.

## 2.4 Water vapour radiometer

On the same tower as the temperature profile, a water vapour radiometer Radiometrics WVR-1100 is installed. It is a microwave receiver measuring the intensity of atmospheric radiation at 23.8 and 31.4 GHz. The water vapour dominates the 23.8 GHz observations, whereas the cloud liquid in the atmosphere dominates the power in the 31.4 GHz channel. This allows the simultaneous determination of integrated water vapour and liquid water along the line of sight. From the

measured brightness temperatures at both frequencies $Tb_{23}$ and $Tb_{31}$, and the blackbody temperature $Tk_{BB}$, the frequency dependent atmospheric opacities $\tau_{23}$ and $\tau_{31}$ are calculated. The water vapour and liquid water content and the phase delay are obtained by using the following relationships:

$$Vap = c0_{vap} + c1_{vap} \cdot \tau_{32} + c2_{vap} \cdot \tau_{31} \tag{1}$$

$$Liq = c0_{liq} + c1_{liq} \cdot \tau_{32} + c2_{liq} \cdot \tau_{31} \tag{2}$$

$$Del = c0_{del} + c1_{del} \cdot \tau_{32} + c2_{del} \cdot \tau_{31} \tag{3}$$

The retrieval coefficients c0, c1 and c2 are site dependent and have to be determined from a history of radiosonde observations from a representative site. The retrieval coefficients used in this work are valid for Munich and displayed in Table 2. A description of the determination of atmospheric water vapour using microwave radiometry is given e.g. in Elgered et al. (1982).






**Table 2. Retrieval coefficients used.**

| $c0_{vap}$ | $c1_{vap}$ | $c2_{vap}$ | $c0_{liq}$ | $c1_{liq}$ | $c2_{liq}$ | $c0_{del}$ | $c1_{del}$ | $c2_{del}$ |
|---|---|---|---|---|---|---|---|---|
| 0.0045 | 23.1680 | -13.9475 | -0.0022 | -0.2705 | 0.5853 | 0.0678 | 151.4489 | -89.7247 |

The instrument performs about 1 measurement per minute in one particular direction. In azimuth steps of 30 degrees,
elevation scans between 20 and 160 degrees are carried out, i.e. the scan passes over the zenith direction. For a complete
scan of the entire sky, it takes about 90 minutes. In order to obtain the zenith delay only, all lines with 90 degree elevation
has to be extracted from the data files. This results in 198 zenith data points per day.

The accuracy of the brightness temperature measurement is specified with 0.5 K. The accuracy of the resulting water vapour
and liquid water contents and phase delays strongly depends on the instrument calibration, i.e. the retrieval coefficients used.

## 2.5 Cloud detector

The cloud detector or nubiscope measures the thermal radiation of the sky in one particular direction. Since clouds absorb
radiation from the sun and reflected infrared radiation from the ground, the temperature of the cloud base is significantly
higher than the blue sky. By scanning the entire sky, a map of the cloud coveraged can be generated. As low clouds
generally yield higher temperatures than high clouds, an additional information regarding the height of the clouds is
obtained. Taking into account the horizon effect, that is the temperature increase from zenith to horizon, the processing
software determines the fraction of low, medium and high level clouds, the coverage, temperature and height of the main
cloud base and the temperature and height of the lowest clouds. Further information is given at the manufacturer's website
(Sattler, no year).

The cloud detector is installed on an observation platform on the roof of the Twin Telescope operation building at 625 m
a.s.l. and 9 m above the surface. The recorded heights of the cloud base refer to the instrument height. A complete scan of
the sky is done once each 10 minutes.

## 2.6 Radiosondes

On every day during the CONT17 experiment, radiosonde balloons were launched at 8:00 and 14:00 UTC at the launch site
depicted in Fig. 1. We used Graw DFM-09 radiosondes and helium filled Totex 350 balloons with 300 g buoyancy. The
transmission rate is one data set per second. The radiosondes are equipped with a GPS receiver permitting an absolute
localization with an accuracy of 5 m in horizontal and 10 m in vertical position. The exact tracking allows precise
measurements of wind speed and wind direction at different heights with an accuracy of 0.2 m/s, and ascent and descent
rates. The air pressure is computed from the surface pressure at the station, the geopotential height and the temperature with
an accuracy of 0.3 hPa. The accuracy of the temperature and relative humidity sensors is specified with 0.2 °C and 4 %,
respectively. The relative humidity $h_{rel}$ can be expressed as water vapour pressure e using





$$e = e_s \cdot \frac{h_{rel}}{100} \tag{4}$$

and the Magnus formula according to Sonntag (1990) for the saturation vapour pressure for water

$$e_s = 6.112 \cdot e^{\frac{17.62 \cdot T}{243.12 + T}} \tag{5}$$

with the temperature T in °C.

Each radiosonde launch yields two files, a profile data file with measured and derived meteorological quantities, and a position data file as coming out from the GPS receiver (see Table 5).

## 3 Weather models

### 3.1 DWD ICON-EU model

For the time span covering the CONT17 campaign, a data set was extracted from the ICON-EU model from the German

Weather Service (Deutscher Wetterdienst, DWD) containing pressure, temperature and humidity data at different height levels. The ICON-EU model is a refined domain (local nest) of the global ICON (**ICO**sahedral **N**onhydrostatic) model, whose grid is made up by a set of nearly equal spherical triangles spanning the entire Earth (Reinert et al., 2018). The ICON-EU nest is refined by dividing each triangle into four subtriangles, resulting in a grid spacing of ~6.5 km. It includes 60 height levels up to 22.5 km. The physical parameters at the top of the model are controlled by the global model reaching a

height of 75 km.

The extracted subset covers a radius of 4 degrees (~445 km) around the GOW. The structure of the grid file 'we_iconeu_4deg.grd' is given in Table 3, where each line represents one of the 13941 grid points. The data files are named 'we_iconeu_4deg_yyyymmddhh.xxx', where yyyy denotes the year, mm the month, dd the day, hh the hour and xxx the physical quantity:

- pre: Air pressure (hPa)
- tem: Temperature (K)
- hum: Water vapour pressure (hPa)

As the model is build up of 60 layers, the temperature and humidity files comprise 60 columns and the pressure file 61 columns, since temperature and humidity is given within the layers and the pressure at the layer boundaries. Each line

represents the same grid point as given in the grid file.



**Table 3.** Structure of the grid file 'we_iconeu_4deg.grd'

| Column: | 1 | 2 | 3 | 4 | ... | 63 |
|---|---|---|---|---|---|---|
| Grid point 1 | Latitude (deg) | Longiude (deg) | Surface (m) | Top Layer 1 (m) | ... | Top Layer 60 (m) |
| ... | ... | ... | ... | ... | ... | ... |
| Grid point 13941 | Latitude (deg) | Longiude (deg) | Surface (m) | Top Layer 1 (m) | ... | Top Layer 60 (m) |

The model data represent the atmospheric analysis fields at the beginning of each forcast run and are computed every 3 hours using assimilated observed data.

**3.2 NCEP model**

As a comparative data set, both zenith hydrostatic and wet delays from the NCEP (National Center for Environmental Prediction) global numerical weather model are provided. This data set is derived from GDAS (Global Data Assimilation System) and GSF (Global Forecasting System) weather fields. The derivation of these tropospheric path delay data requires some explanation, because only one dimensional output files from the GDAS numerical weather model (so-called "surfaces

fluxes") were used. From our experience, zenith total delays are expected to reveal a standard deviation approaching one centimetre for the region of Wettzell. This is slightly less accurate than the estimation of tropospheric delays using GNSS permanent stations (see Fig. 10), but still useful for a number of applications.

The original weather model output data can be found on ftp server *ftpprd.ncep.noaa.gov* in directory */pub/data/nccf/com/ gfs/prod*, all available in standard grib2 format. Note that this is a rolling real-time archive. Regions of interest are routinely

extracted at our observatory and converted into a cut-tailored format addressing the specific needs of space geodesy. Analysis fields are used whenever possible (every 6 hours) with one 3 hour prediction in between.

The needed information is horizontally interpolated and vertically reduced to the central GNSS station WTZR at the observatory. The horizontal interpolation approach is depicted in (Schüler, 2001, p. 197ff) using the 4 nearest neighbours, but as a modification, bi-linear functions of type $a_0 + a_1 \cdot \varphi + a_2 \cdot \lambda + a_3 \cdot \varphi \cdot \lambda$ are employed for interpolation of the surface flux

data, where $a_{0..3}$ are the interpolation coefficients determined from the 4 nearest neighbours, $\varphi$ is the latitude of the interpolation site, and $\lambda$ is its longitude. Vertical reduction to the target height is important. The TropGrid2 model (Schüler, 2014) is used for this purpose. TropGrid2 is a global gridded 1° x 1° model containing reduction coefficients for all quantities needed. The coefficients of these reduction functions were derived using 9 years of numerical weather model data. The determination of *ZHD* (zenith hydrostatic delay) from GDAS/GSF surface fields is straightforward: Surface pressure is

horizontally interpolated and vertically reduced, and then converted into *ZHD* using the Saastanoinen model (Saastamoinen, 1972)

$$ZHD = \frac{0.0022767 \cdot p}{1 - 0.00266 \cdot \cos(2\varphi) - 0.00028 \cdot h} \tag{6}$$





with the pressure $p$ (hPa), the ellipsoidal height $h$ (km) and the geographic latitude $\varphi$ of the station. The derivation of *ZWD* (zenith wet delay) requires more effort, but GDAS/GSF surface fluxes are a very attractive resource since these weather fields already contain the total column atmospheric water vapour (*IWV*, integrated water vapour). These values are converted into *ZWD* with knowledge of the weighted mean temperature of the atmosphere $T_M$ (see Schüler, 2001, p. 184ff). $T_M$ itself is

substituted in the standard product by a surface temperature conversion function available on the TropGrid2 data grid. After conversion, ZWD is vertically reduced and horizontally interpolated to the target height.

**4 Data representation and results**

The data from the **radiosonde balloon ascents** give a direct temperature and humidity profile through the troposphere and are thus a proper tool to validate the wheather model and to calibrate radiation based sensors like the water vapour

radiometer or the temperature profiler. The radiosonde ascents between Nov. 28 and Dec. 15 reached heights between 5,6 km (2017121408) and 25,8 km (2017121308) with an average at 19 km. The average ascent rates were between 4 and 6 m/s in most cases. The maximum covered horizontal distance to the burst point was 170 km towards Northeast (Fig. 2). The horizontal drift is 2-8 km per km height in most cases (Fig. 3). This means that the tropospheric data up to 10 km height is representative for a region 20-80 km around the launch site.

The comparison of the radiosonde temperature profiles with those of the weather model show a mostly very good agreement. Some small scale perturbations in the radiosonde date are not present in the model, however, the trend is always in accord. The linear regression between the weather model temperatures and those from the radiosondes being interpolated to the model layer heights yield linear trends (m) and correlation coefficients (cc) being very close to 1 (see example in Fig. 4) underlining the high consistency of the model. The only misfit occured at the 2017120108 launch. In this particular case the

measured height seemed to be corrupted.

A slightly worse agreement exists between the water vapour contents of the weather model and those derived from the radiosonde measurements. As for the temperature small scale perturbations are not represented in the weather model. The general trend is similar, however, the model tends towards higher water vapour contents, which is also expressed in the greater slope of the trend line (Fig. 5), which are between 1.0 and 1.2 in most cases. The correlation is good with cc usually

greater than 0.98.

A graphical representation of measured pressure, temperature and water vapour profiles from all radiosonde ascents in comparison to model data is given in the supplement.





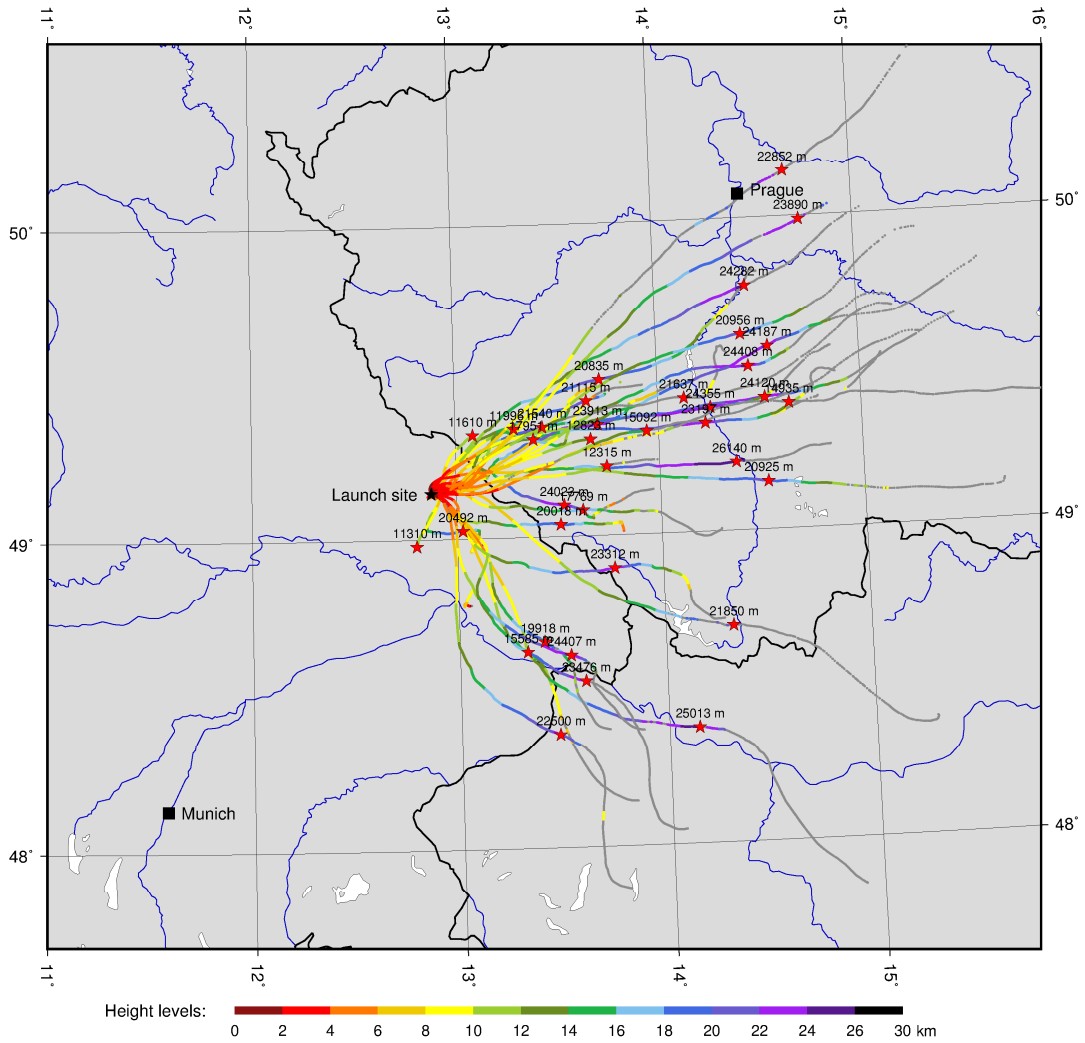

**Figure 2: Traces of radiosonde balloons with maximum heights indicated by red stars.**





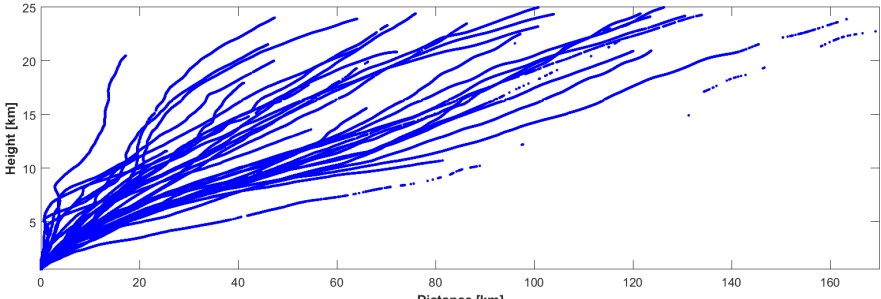

**Figure 3: Height-distance plot of all balloon ascents. Height axis is exaggerated by factor 2.**

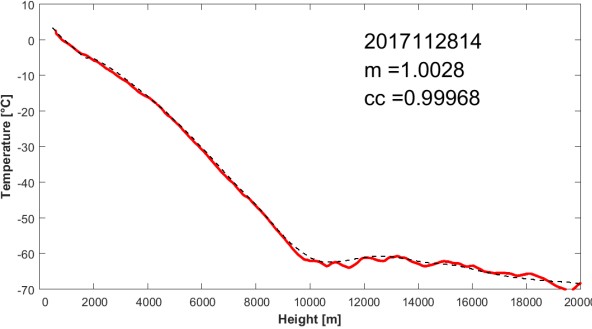

**Figure 4: Temperature vs. height profile of one particular radiosonde ascent (red line) as compared to the weather model profile at the launch location (dotted line). The correlation parameters between both series (m: slope of the best fit line, cc: correlation coefficient) are indicated.**

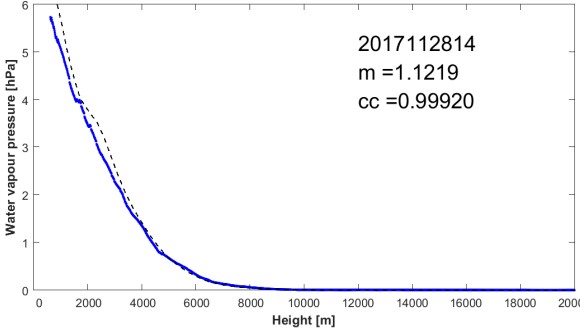

**Figure 5: Water vapour content vs. height profile of one particular radiosonde ascent (blue line) as compared to the weather model profile at the launch location (dotted line). The correlation parameters between both series (m: slope of the best fit line, cc: correlation coefficient) are indicated.**



The radiosonde data can also be used to validate the **temperature profiler**. Figure 6 shows the traces of the temperature profiler at 6 different height levels compared to temperatures measured by the radiosondes in the equivalent height. While a good coincidence is given at heights up to 400 m, the higher levels yield systematically higher temperatures using the

5   profiler. The rms of the temperature differences (rms_dif) at a particular height increases from 0.82 at 250 m up to 2.14 at 1000 m (Table 4). This behaviour is underlined by the parameters of linear regression between both temperatures. The slope of the regression line (b) is always lower than 1 and the y-axis offset (a) increases with height. This indicates that the profiler underestimates particularly the lower temperatures at higher levels. Examples for a good and a poor coincidence are given in Fig. 7.

**Table 4. Parameters from linear regression between temperatures from radiosonde ascents (x) and temperature profiler (y): slope b, y-axis offset a, rms fit error and rms of temperature differences.**

| Height | b | a | rmse | rms_dif |
|---|---|---|---|---|
| [m] | | [°C] | [°C] | [°C] |
| 0 | 0.825 | -0.215 | 1.128 | 1.488 |
| 50 | 0.833 | 0.362 | 1.065 | 1.342 |
| 100 | 0.866 | 0.468 | 0.843 | 1.126 |
| 150 | 0.902 | 0.419 | 0.720 | 0.942 |
| 200 | 0.932 | 0.518 | 0.587 | 0.857 |
| 250 | 0.933 | 0.434 | 0.592 | 0.821 |
| 300 | 0.927 | 0.410 | 0.636 | 0.869 |
| 350 | 0.927 | 0.274 | 0.761 | 0.912 |
| 400 | 0.928 | 0.277 | 0.823 | 0.975 |
| 450 | 0.934 | 0.179 | 0.865 | 0.970 |
| 500 | 0.932 | 0.121 | 0.913 | 1.009 |
| 550 | 0.930 | 0.101 | 0.997 | 1.089 |
| 600 | 0.932 | 0.193 | 1.083 | 1.197 |
| 650 | 0.932 | 0.303 | 1.160 | 1.308 |
| 700 | 0.935 | 0.545 | 1.238 | 1.483 |
| 750 | 0.932 | 0.679 | 1.280 | 1.607 |
| 800 | 0.931 | 0.898 | 1.248 | 1.734 |
| 850 | 0.927 | 0.975 | 1.233 | 1.800 |
| 900 | 0.921 | 1.137 | 1.235 | 1.959 |
| 950 | 0.913 | 1.173 | 1.229 | 2.033 |
| 1000 | 0.907 | 1.270 | 1.193 | 2.141 |



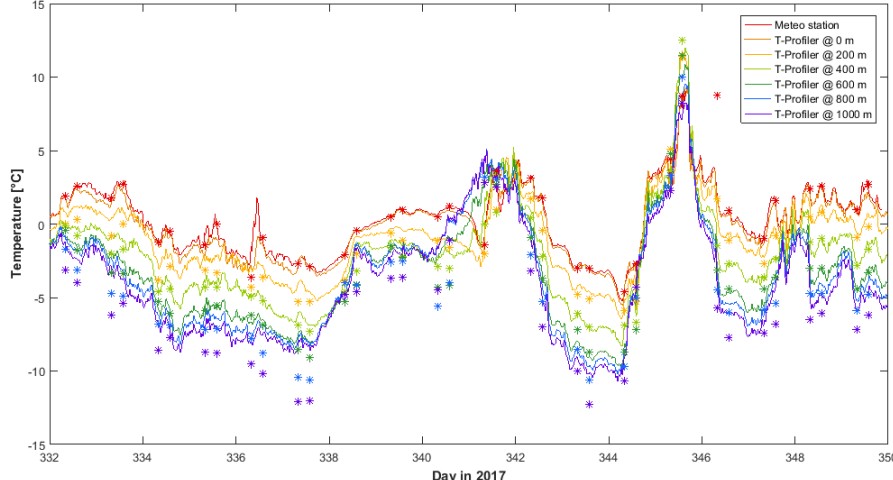

**Figure 6: Temperature profiler time series at particular heights compared to temperature record of meteo station and radiosonde data at equivalent heights (asterisks).**

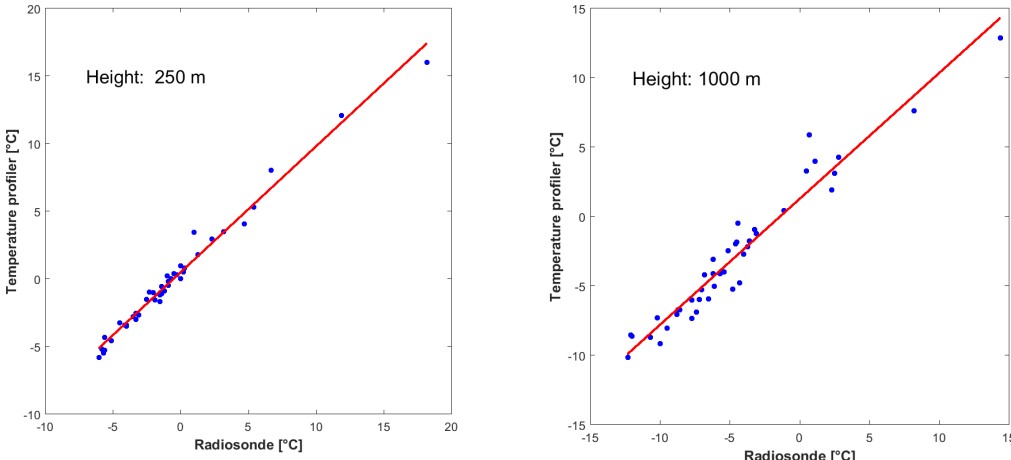

**Figure 7: Linear regression between temperatures from radiosonde ascents and contemporaneous profiler records at two particular heights. Regression parameters see Table 4.**



One quantity measured by the **water vapour radiometer** is the integrated water vapour content given in height of the eqivalent water column. In order to compare this quantity with weather model and radiosonde data, the water vapour pressure $e$ was converted to specific humidity $s$ using the following relationship:

$$5 \quad s = \frac{0.622 \cdot e}{p - 0.378 \cdot e} \tag{7}$$

The dimensionless parameter $s$ is then integrated level by level over the vertical column of the weather model or radiosonde profile, respectively. The resulting water height equivalents are compared with those measured by the WVR in Fig. 8. The general agreement is good, however, the WVR produces outlieres during periods of rain. This known issue is a consequence of rain droplets resting on the radiometer window and falsifying the results. The linear regression (Fig. 9) shows a slightly

better agreement between the radiosonde and the weather model than between the radiosonde and the WVR, which tends to slightly overestimate the water vapour content. It should be noted, however, that the retrieval coefficients used here are valid for Munich, which is 200 km away. In addition, the vertical profile of the radiosonde is not necessarily representative for the launch site due to the horizontal drift of the balloon (see Fig. 3).

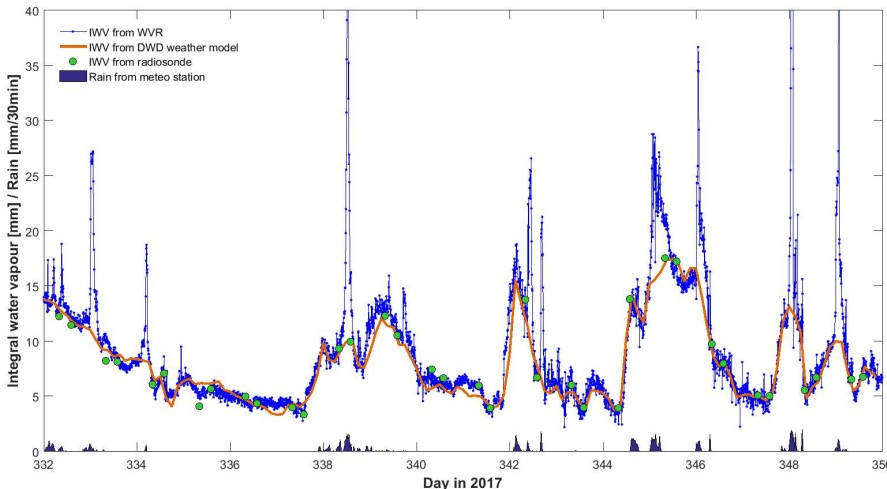

**Figure 8: Integrated water vapour (IWV) content as measured by the water vapour radiometer (WVR) compared to IWV values derived from weather model and radiosonde data. WVR spikes coincide with periods of rain.**




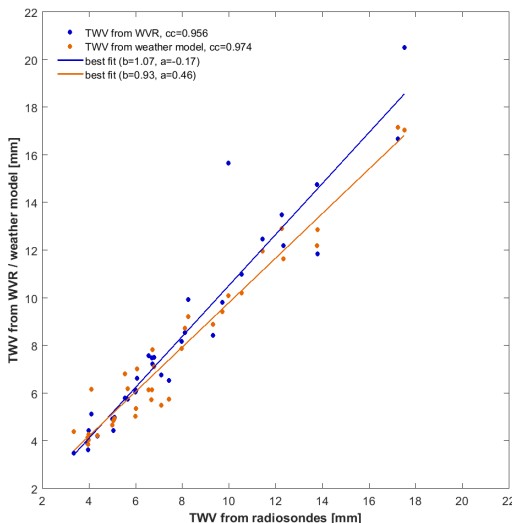

**Figure 9: Linear regression between integrated water vapour (IWV) content derived from radiosonde data and those taken from water vapour radiometer (WVR) and weather model data.**

The water content is an important quantity for the estimation of the **Zenith Total Delay (ZTD)**, that is the delay radiowaves undergo during their propagation through the atmosphere. The zenith delays can be mapped to the slant path by using geometric relationships, e.g. the Niell mapping function (Niell, 1996) or the Vienna mapping function (Böhm et al., 2006). The ZTD can be split into a dry, hydrostatic part (Zenith Hydrostatic Delay, ZHD) and a wet part (Zenith Wet Delay, ZWD). Both zenith delay components are obtained by vertical integration of the refractivity indices $N_{hyd}$ and $N_{wet}$ for each model

layer over the entire model. The hydrostatic refractivity index $N_{hyd}$ depends only on the air density $\rho$

$$N_{hyd} = k_1 \cdot R_d \cdot \rho \tag{8}$$

with the hydrostatic refraction constant $k_1 = 77.6$ K hPa$^{-1}$ and the specific gas constant for dry air $R_d = 287.05$ J kg$^{-1}$ K$^{-1}$. The density follows the equation of state for ideal gases

$$\rho = \frac{p}{R_d \cdot T_v} \tag{9}$$

with the pressure p and the virtual temperature $T_v$ in each layer. $T_v$ is the equivalent temperature of dry air having the same density as wet air, and is computed from the air temperature T and the specific humidity s according to (Emeis, 2000):

$$T_v = T \cdot (1 + 0.608 \cdot s) \tag{10}$$



The wet refractivity index $N_{wet}$ is a function of the partial water vapour pressure e and the temperature T in Kelvin

$$N_{wet} = k_2' \cdot \frac{e}{T} + k_3 \cdot \frac{e}{T^2} \qquad (11)$$

with the refraction constants $k_2'$ = 22.1 K hPa$^{-1}$ and $k_3$ = 370100 K$^2$ hPa$^{-1}$ (Bevis, 1994). The compressibility factor accounting for non-ideal gas behaviour is neglected in this case.

For the vertical integration, the refractive index at each layer times the layer thickness is summed up over all model layers. Above the upper boundary of the ICON-EU model at 22.5 km height, the remaining part of ZHD, being in the order of 7-8 cm, is computed acoording to Eq. (6) with the pressure and height taken at the top of the model instead of the surface. The contribution of the atmosphere above 22.5 km to the ZWD can be neglected since the water vapour content is close to zero. A similar procedure was applied to determine the zenith delays ZHD and ZWD from radiosonde data.

The total delays ZTD being the sum of ZHD and ZWD as computed from weather model and radiosonde data are displayed in Fig. 10 and compared to the ZTD estimation from GNSS analyses. One solution is taken from the BKG GNSS data center, a routine analysis of station WTZR as part of the of the GREF network *(https://igs.bkg.bund.de/dataandproducts/browse)* using Bernese 5.2 software, the other solution is derived from the Wettzell local array using the inhouse analysis software SGSS. The reported values represent the mean and the 68 % confidence interval of the 8 Wettzell GNSS stations each being

analyzed in three different regional networks. The confindence intervals give a more realistic error estimation and are thus larger than the standard deviations of a single analysis given in the GREF data.

A time series of the different ZTD values is displayed in Fig. 11. All traces show a similar behaviour. The GNSS analyses reveal more details as a consequence of the higher sampling rate of 1 hour. Taking the radiosonde data as a reference, the DWD model tends towards lower (2-3 mm) and the NCEP model towards higher (5-6 mm) ZTD values. The best

coincidence with the radiosonde derived ZTD give the GNSS solutions with correlation coefficients up to 0.992.





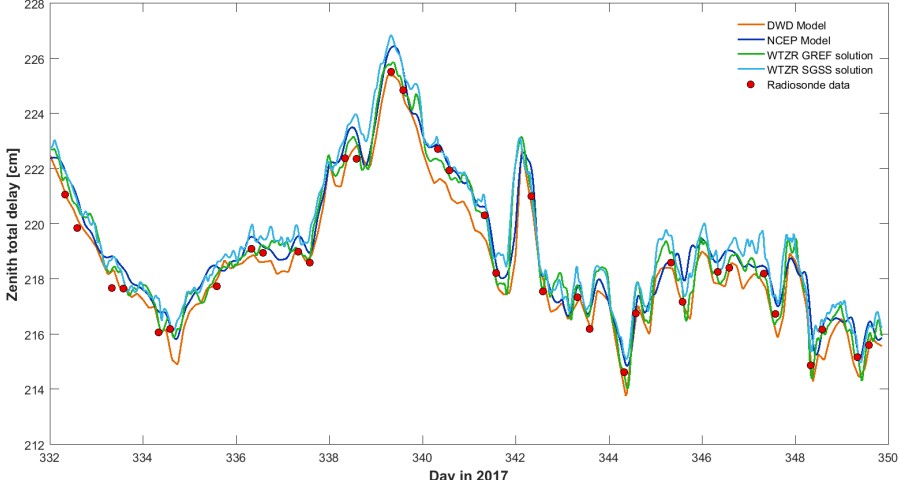

**Figure 10: Zenith total delays (ZTD) derived from numerical weather models, GNSS solutions and radiosonde data.**

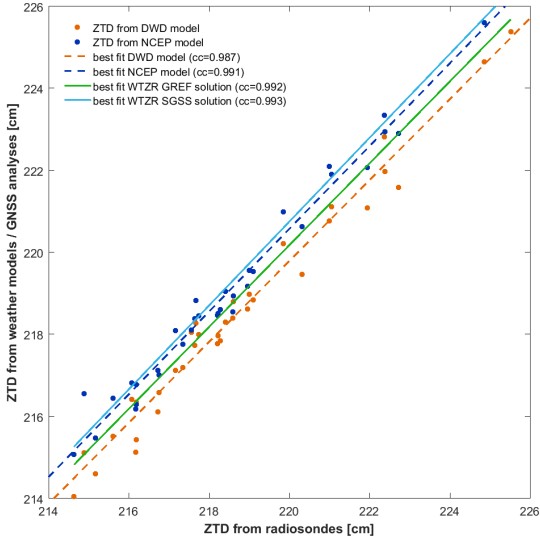

**Figure 11: Linear regression between zenith total delays (ZTD) derived from radiosonde data and those derived from numerical weather models and GNSS solutions.**




The cloud coverage as recorded by the **nubiscope** and the global radiation as measured by the **pyranometer** are displayed in Fig. 12.

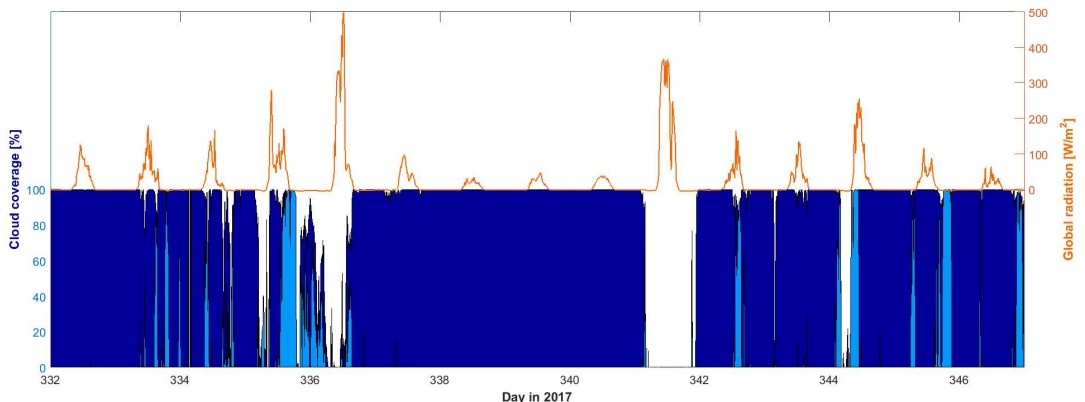

5 **Figure 12: Total cloud coverage (dark blue) and portion of medium plus high level clouds (light blue) in comparison with the global radiation as measured by the pyranometer.**

## 5 Data access and file structure

All data sets are available at https://doi.pangaea.de/10.1594/PANGAEA.895518. In all time series, the first column represents UTC date and time having the format yyyy-mm-ddThh:mm:ss. The colums are separated by tabs (\t) in all files
10 with the exception of the ICON-EU model data where blanks (\s) are used. The ICON-EU model data are stored in a compressed tar archive, all other files are available as ASCII text files. The file description is given in Table 5.

**Table 5. Description of the dataset.**

| Dataset | File | Content |
|---|---|---|
| Meteorological observations | CONT-17_Wettzell_meteo.tab | See Table 1 |
| Global and net radiation | CONT-17_Wettzell_rad.tab | Short-wave downward (global) radiation and net radiation (W/m$^2$) |
| Temperature profile | CONT-17_Wettzell_Tpro.tab | Radiometric temperatures (°C) between 0 and 1000 m above ground, ambient temperature in the last column |
| Water vapour and liquid water | CONT-17_Wettzell_vapo.txt | Water vapour radiometer data: Tb23, Tb31: brightness temperatures (K), TkBB: blackbody temperature (K), VapCM, LiqCM: integrated |





| | | |
|---|---|---|
| content | | water vapour and liquid water content (cm water column), DelCM: radiometric delay (cm), AZ, EL: azimuth and elevation (°), Tau23, Tau31: atmospheric opacities, T_amb: ambient temperature (°C), RH: relative humidity (%), P: pressure (hPa), Rain: rain identifier (arbitrary unit) |
| Cloud coverage and cloud temperatures | CONT-17_Wettzell_nubi.txt | Pr: Precipitation flag, Tgrnd: ground temperature (°C), Tbase: model base temperature (°C), Tzero: air temperature (°C), Tblue: infrared temperature of clear sky at zenith (°C), Type (Clear Sky, Cirrus Only, Broken Clouds, OverCast, Transparent Clouds, Low Transparent clouds, Fog, Reduced Visibility), ClCov: total cloud coverage (%), <MCB: clouds below main cloud base (%), MCB: coverage (%) base temperature (°C) and height (m) of main cloud base, LLC: coverage of low level clouds (%), MLC: coverage of medium level clouds (%), HLC: coverage of heigh level clouds (%), lowestCl: base temperature (°C) and height (m) of lowest clouds |
| Radiosonde data | CONT-17_Wettzell_radios.tab | Sonde ID, time (s after launch), latitude (°), longitude (°), altitude (m), pressure (hPa), temperature (°C), relative humidity (%), wind speed (m/s), wind direction (° cw from North), geopotential height (m) |
| ICON-EU model data | iconeu_wtz.grd | Latitude (°), longitude (°) and height levels (m) (see Table 3) |
| | iconeu_wtz_yyyymmddhh.pre | Air pressure (hPa) at layer boundaries (see Sect. 3.1) |
| | iconeu_wtz_yyyymmddhh.tem | Temperature (K) within layers (see Sect. 3.1) |
| | iconeu_wtz_yyyymmddhh.hum | Water vapour pressure (hPa) within layers (see Sect. 3.1) |
| NCEP model data and zenith path delays | CONT-17_Wettzell_ncep-sflux-zpd.tab | Surface fluxes from NCEP model and derived zenith path delays (see Sect. 3.2) interpolated to WTZR location: Air pressure (hPa), temperature (°C), relative humidity (%), zonal and meriodonal wind speed (m/s), cloud coverage (%), precipitation rate (mm/h), weighted mean temperature (°C), Zenith total delay ZTD (mm), zenith hydrostatic delay ZHD (mm) and zenith wet delay ZWD (mm) with standard deviations SD |
| Zenith path delays from GNSS analysis | CONT-17_Wettzell_zpd_sgss_gref.tab | ZTD (mm) from local network analysis using SGSS software with 68% confidence interval C of median value, ZTD (mm) from GREF analysis with standard deviations |





**Author contribution**

TS initiated the project and the radiosonde balloon ascends, which were performed under supervision of WS. AB and WS maintained the instruments and provided the measured data. Model data were prepared by TS and TK. TK compiled the data and prepared the manuscript with contributions from all co-authors.

**Competing interests**

The authors declare that they have no conflict of interest.

**Acknowledgements**

The support from the entire team of the Geodetic Observatory Wettzell is gratefully acknowledged.

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
