# Peer review of "Atmospheric data set from the Geodetic Observatory Wettzell during the CONT-17 VLBI campaign"

_Earth System Science Data, 2018_

## Referee Comment (RC1) · Anonymous Referee #1 · 11 Dec 2018

General Comments

Overall I find that the dataset is of general interest to the space geodetic community. I think that 15 days of data is a too small dataset to be of interest to the meteorological community, given that it does not coincide with an extreme weather event.

One issue which I find difficult to judge concerns the following instruction to reviewers:

"Completeness: A data set or collection must not be split intentionally, for example, to increase the possible number of publications. It should contain all data that can be reviewed without unnecessary increase of workload and can be reused in another context by a reader."

This submitted dataset is acquired at the Wettzell VLBI station during the CONT17

experiment. There were approximately 30 stations participating in the different networks of the CONT17 experiment, meaning that a more complete dataset could include weather data from all these stations. From the geodetic point of view such a data set would be much more useful. Are the authors aware of if there are plans to publish similar data sets also from other stations? Ideally all data would be published as one dataset, although I realise that it may be a huge work load thinking about the logistics and that the atmospheric measurements are (probably) not a main focus at many stations.

An alternative data set would be to combine the atmospheric data from many CONT experiments at Wettzell. Wettzell has participated in CONT02, CONT05, CONT08, CONT11, and CONT14, see e.g. Teke et al. (2013) and Lu et al. (2017). Of course, a dataset including the available atmospheric data from all these experiments would be more "complete". On the other hand it is foreseen that this type of CONT experiments will continue to be carried out every three years.

The question is if it is meaningful to produce one additional atmospheric paper/dataset corresponding to each site and each experiment? I believe this is mainly an editorial question. In any case the dataset would benefit from describing it in this broader context. For example, among other things, include the references:

Lu, Cuixian, Xingxing Li, Maorong Ge, Robert Heinkelmann, Tobias Nilsson, Benedikt Soja, Galina Dick, Harald Schuh (2016). Estimation and evaluation of real-time precipitable water vapor from GLONASS and GPS, GPS Solut., 20:703–713, DOI 10.1007/s10291-015-0479-8.

Teke, Kamil, Tobias Nilsson, Johannes Böhm, Thomas Hobiger, Peter Steigenberger, Susana García-Espada, Rüdiger Haas, Pascal Willis (2013). Troposphere delays from space geodetic techniques, water vapor radiometers, and numerical weather models over a series of continuous VLBI campaigns, J. Geod., 87:981–1001 DOI 10.1007/s00190-013-0662-z.

Specific comments

Concerning the definition of atmospheric observations I note that when describing the instrumentation (in Section 2) you chose to neglect the GNSS and the VLBI observations. But in Section 4 you present the estimated ZTD from the GNSS observations, whereas estimated ZTDs from the VLBI observations are not included at all. I find this confusing because the space geodetic techniques have the advantage of not being based on emission measurements, possibly having the potential of producing stable long term time series for the ZTD (and indirectly the water vapour content). The bottom line is that VLBI and GNSS may be used to assess the absolute accuracy of the other meteorological sensors and should be described and included in the dataset.

I find Table 1 difficult to interpret. Each parameter is listed with an accuracy, but the accuracy is not defined (absolute traceable to SI, one standard deviation, or two, or three standard deviations?). The parameter SM is not shown in the map of Figure 1. The accuracy of R1 and R2 is stated to be 10 %. Precipitation is not measured in percentage. What is the unit that has this relative uncertainty? I also think that the table will be more clear if the 3rd line would be on the top (title line) and the acronyms on the line below.

The question of defining accuracy is also valid for the brightness temperatures measured by the radiometer (page 6, line 8). Is 0.5 K an absolute accuracy or x standard deviations?

On page 9 you refer to good agreement twice (lines 15 and 24). The word good is a rather subjective statement and have different meanings to different persons. I think it shall be avoided and instead specify the quality of the agreement in numbers, such as RMS differences and correlation coefficients.

You comment on that radiometer data are more or less useless to infer water vapour content, liquid water content, and wet delay during rain. Actually, also when large drops of liquid water are present in the sensed volume of air, the accuracy will be worse. A

similar effect is if water drops are present on the feed/mirror of the radiometer, which will often be the case for some time also after that the rain has stopped. In Figure 9 you have two outliers (blue dots). It may be worthwhile to investigate if these are in connection to a rain shower or large amounts of the liquid water content.

A relevant question for this type of (data) manuscript is how far it is reasonable to take the data analysis? Had it been a regular scientific paper I would have argued that instead of just using retrieval coefficients for the radiometer data from radiosonde data obtained in Munich, it would be required to at least also compare these coefficients from the ones that can be obtained from the launches carried out at the Wettzell site. On the other hand, one reason for publishing a dataset is to inspire others to use it. This could be one such task.

On page 5, line 20 and in Table 5 you use the parameter Tk_{BB} referred to as black-body temperature which is not defined. Given that it in the dataset is about 10 K warmer than the ambient temperature it cannot be the effective temperature of the atmosphere that is used to calculate the optical depth at the observed frequencies (which also is frequency dependent)?

The dataset (described in Table 5) should, where possible, have an uncertainty attached to each parameter. For example, uncertainties in the observed sky brightness temperatures propagate and give, together with uncertainties in the retrieval coefficients, uncertainties in the inferred parameters.

Technical Corrections

I find that the font size in all figures is unnecessarily small. The size could in general be say 30-50 % larger in order to improve the readability.

page 2, lines 5-6: The weather model data are not acquired at Wettzell. Suggest to rewrite as: ... atmospheric observations were acquired at the Geodetic Observatory Wettzell, where three radio telescopes contributed to three different networks which

have been established for this campaign. These data were supplemented by weather model data. The data set is made ...

page 2, line 22: Referring to the IVS home page the appropriate general reference to the IVS is: A. Nothnagel, T. Artz, D. Behrend, Z. Malkin, "International VLBI Service for Geodesy and Astrometry – Delivering high-quality products and embarking on observations of the next generation", Journal of Geodesy, Vol. 91(7), pp. 711–721, July 2017. DOI 10.1007/s00190-016-0950-5

page 2, line 10: all –> many

page 2, line 13: -temperatures –> cloud temperatures

page 2, line 16: resolution –> resolutions + is very different –> are very different

page 3, line 1: radiotelescopes –> radio telescopes

page 4, line 4: humidity –> humidity,

page 5, line 21: phase delay –> path delay (I guess it is expressed in units of length and not in degrees since the phase delay depends on the carrier frequency)

page 6, Table 2: units are missing

page 6, line 13: blue –> clear ?

page 6, line 18: no year –> (2018) (according to the reference list)

page 6, line 26: delete "exact" (which means without error and that cannot be true)

page 7, line 2: for water –> for water in hPa ?

page 9, line 14: around the launch site –> mainly to the east of the launch site

page 9, line 24: define or write out cc ?

page 9, line 26: temperature –> temperature,

page 11, Figures 4 and 5: RMS difference is a better parameter compared to cc to describe the quality of the agreement because it is not so strongly depending on the dynamic range of the measured values. Also you may refer to Table 4 in the figure captions? I am used to see height on the x-axis and the measured meteorological parameter on the y-axis in this type of meteorological graphs.

page 12, line 26: for a good and a poor coincidence –> of one better and one worse agreement

page 12, Table 4: write out "RMS error" and "RMS difference" ?

page 13, Figure 7: add also the ideal line for a perfect agreement ?

page 13, line 5: waht is the unit of s ? Reference?

page 14, line 1: measured –> inferred from the measured sky brightness temperatures

page 15, line 16: (Emeis, 2000) is not in the list of references

page 16, line 3: Bevis –> Bevis et al.

page 17, Figure 11: It would be informative to add values for the RMS differences in addition to the correlation coefficients.

page 19, Table 5: 68% –> 68 %

page 20, line 16: J. Applied Meteorology –> J. Appl. Meteorol.

page 20, line 21: Elegered –> Elgered

page 20, line 22: Radio Science –> Radio Sci.

---

## Referee Comment (RC2) · Anonymous Referee #2 · 13 Dec 2018

Comment on the manuscript by Thomas Klügel, Armin Böer, Torben Schüler, and Walter Schwarz: Atmospheric data set from the Geodetic Observatory Wettzell during the CONT-17 VLBI campaign.

CONT-17 is the most recent continuous VLBI campaign over two weeks organized by the International VLBI Service for Geodesy and Astrometry (IVS) to assess and push the frontiers of current geodetic VLBI capabilities. For example, it is the ideal test bed to determine high-resolution Earth rotation parameters and other geodetic quantities from three different networks (A, B, and VGOS). One very important error source in VLBI is the modelling of tropospheric delays. Consequently, CONT-17 is perfectly suited to assess the modelled and estimated tropospheric delays at the participating sites, e.g. by comparison with other techniques like GNSS, water vapor radiometers or numerical

weather models. In the past, it has always been rather difficult and cumbersome to collect information from other sources. Here, the authors provide a unique data set to the scientific community, which can be used for many studies related to the geodetic observatory in Wettzell and CONT-17 in general. In the following, I am going to highlight a few of those: The data set, in particular the radiosonde data but also the weather modes, are well suited to derive the best possible models like mapping functions. These mapping functions can then be used to validate existing models like the Vienna Mapping Functions. Moreover, locally measured meteorological data are very useful for the determination of local atmospheric ties. The combination of the various data sets can be used to derive information about turbulence, etc.

The manuscript is very clear and well written. I randomly checked the provided datasets on Pangaea, and I could well assess the content. I very much appreciate the possibility to see the data in html and to plot time series. Thanks to the authors and the team at the Geodetic Observatory Wettzell for providing this special and unique dataset! The scientific community will certainly use the data.

I just found two typos on page 16: contrubution, confindence

---

## Author Comment (AC1) · 9 Jan 2019

The authors highly appreciate the comments of the referees. They were very helpful to improve the manuscript.

Reply to general comments:

It is generally a good idea to publish atmospheric data sets from other stations having participated in the CONT 17 experiment. Basic information like air pressure, temperature or humidity is contained in the VLBI data files. Additional atmospheric information is usually not provided. The main goal of this paper is to provide an outstanding atmospheric data set including vertical information and model data to have an optimal basis to compare different techniques and to establish a consistent common atmosphere

being valid for all geodetic space techniques operated in Wettzell. For that reason atmospheric data from other stations were not included.

If stored on the same server, data sets from other stations could be linked to the same campaign ("CONT-17") as a different station/event.

During the former CONT experiments, only the standard meteorological ground observations plus WVR data were acquired in Wettzell. Indeed, such a data set would be more complete, however, due to the lack of auxiliary data like radiosonde ascents, the meteorological data sets from former experiments do not contain substantial new information. It could be beneficial to provide similar extensive meteorological data sets for future CONT campaigns, also from other VLBI stations. The authors explicitly support this idea.

Reply to specific comments:

In the context of ZTD determination, the inclusion of GNSS and VLBI analysis data is of high interest. However, since the goal is to provide a sound data base mainly for VLBI analysists, the presentation and description of analysis results and procedures would break the scope of this paper. The presentation of GNSS tropospheric delays is only for the purpose of comparison. As a continuously generated routine product, the analysis procedure for the GNSS ZTD estimation has not been described here.

Table 1 has been changed according to the reviewer's suggestions. The accuracy is a manufacturer information and can't be specified more in detail. The same is true for the given accuracy of the brightness temperature of the WVR.

The good correlation between radiosonde and model data (page 9) is underlined by explicitly mentioning the mean correlation coefficient.

The 2 outliers in fig. 9 are caused by a previous rain event. The text and fig. 9 were adopted accordingly.

Regarding the use of retrieval coefficients from Munich, we actually intended to derive

own retrieval coefficients being valid for the Wettzell site. However, the WVR manufacterer stated that a reliable determination of retrieval coefficients requires continuous radiosonde data over at least 1 year, which were not available at our site. So we chose to use a coefficient set from a neighboring site.

The parameter Tk_BB (blackbody temperature) is now explained in the text.

Regarding the uncertainties of the inferred WVR-parameters, the following sentence has been added on page 14: "Thus the total accuracy of the estimated water vapour and liquid water content, where uncertainties from the brightness temperature measurement and retrieval coefficients sum up, can't be specified."

Technical corrections:

All corrections mentioned by the referee were applied.

---

## Author Comment (AC2) · 16 Jan 2019

The authors are grateful for the comments of the referee.

All corrections mentioned by the referee were applied.

—————————————————

---

## Author Response (AR1)

**Answers to comments of Referee #1**

General comments

5   Comment from Referee:

Overall I find that the dataset is of general interest to the space geodetic community. I think that 15 days of data is a too small dataset to be of interest to the meteorological community, given that it does not coincide with an extreme weather event.

One issue which I find difficult to judge concerns the following instruction to reviewers:

10   "Completeness: A data set or collection must not be split intentionally, for example, to increase the possible number of publications. It should contain all data that can be reviewed without unnecessary increase of workload and can be reused in another context by a reader."

This submitted dataset is acquired at the Wettzell VLBI station during the CONT17 experiment. There were approximately 30 stations participating in the different networks of the CONT17 experiment, meaning that a more complete dataset could

15   include weather data from all these stations. From the geodetic point of view such a data set would be much more useful. Are the authors aware of if there are plans to publish similar data sets also from other stations? Ideally all data would be published as one dataset, although I realise that it may be a huge work load thinking about the logistics and that the atmospheric measurements are (probably) not a main focus at many stations.

20   Author's response:

It is generally a good idea to publish atmospheric data sets from other stations having participated in the CONT 17 experiment. Basic information like air pressure, temperature or humidity is contained in the VLBI data files. Additional atmospheric information is usually not provided. The main goal of this paper is to provide an outstanding atmospheric data set including vertical information and model data to have an optimal basis to compare different techniques and to establish a

25   consistent common atmosphere being valid for all geodetic space techniques operated in Wettzell. For that reason atmospheric data from other stations were not included.

Comment from Referee:

An alternative data set would be to combine the atmospheric data from many CONT experiments at Wettzell. Wettzell has

30   participated in CONT02, CONT05, CONT08, CONT11, and CONT14, see e.g. Teke et al. (2013) and Lu et al. (2017). Of course, a dataset including the available atmospheric data from all these experiments would be more "complete". On the other hand it is foreseen that this type of CONT experiments will continue to be carried out every three years.

Author's response:

During the former CONT experiments, only the standard meteorological ground observations plus WVR data were acquired in Wettzell. Indeed, such a data set would be more complete, however, due to the lack of auxiliary data like radiosonde ascents, the meteorological data sets from former experiments do not contain substantial new information. It could be beneficial to provide similar extensive meteorological data sets for future CONT campaigns, also from other VLBI stations. The authors explicitly support this idea.

Comment from Referee:

The question is if it is meaningful to produce one additional atmospheric paper/dataset corresponding to each site and each experiment? I believe this is mainly an editorial question.

Author's response:

If stored on the same server, data sets from other stations could be linked to the same campaign ("CONT-17") as a different station/event.

Comment from Referee:

In any case the dataset would benefit from describing it in this broader context. For example, among other things, include the references:

Lu, Cuixian, Xingxing Li, Maorong Ge, Robert Heinkelmann, Tobias Nilsson, Benedikt Soja, Galina Dick, Harald Schuh (2016). Estimation and evaluation of real-time precipitable water vapor from GLONASS and GPS, GPS Solut., 20:703–713, DOI 10.1007/s10291-015-0479-8.

Teke, Kamil, Tobias Nilsson, Johannes Böhm, Thomas Hobiger, Peter Steigenberger, Susana García-Espada, Rüdiger Haas, Pascal Willis (2013). Troposphere delays from space geodetic techniques, water vapor radiometers, and numerical weather models over a series of continuous VLBI campaigns, J. Geod., 87:981–1001, DOI 10.1007/s00190-013-0662-z.

Author's response:

A corresponding sentence and the two references were added in the introduction.

Specific comments

Comment from Referee:

Concerning the defnition of atmospheric observations I note that when describing the instrumentation (in Section 2) you chose to neglect the GNSS and the VLBI observations. But in Section 4 you present the estimated ZTD from the GNSS observations, whereas estimated ZTDs from the VLBI observations are not included at all. I find this confusing because the

space geodetic techniques have the advantage of not being based on emission measurements, possibly having the potential of producing stable long term time series for the ZTD (and indirectly the water vapour content). The bottom line is that VLBI and GNSS may be used to assess the absolute accuracy of the other meteorological sensors and should be described and included in the dataset.

Author's response:

In the context of ZTD determination, the inclusion of GNSS and VLBI analysis data is of high interest. However, since the goal is to provide a sound data base mainly for VLBI analysists, the presentation and description of analysis results and procedures would break the scope of this paper. The presentation of GNSS tropospheric delays is only for the purpose of
10    comparison. As a continuously generated routine product, the analysis procedure for the GNSS ZTD estimation has not been described here.

Comment from Referee:

I find Table 1 difficult to interpret. Each parameter is listed with an accuracy, but the accuracy is not defined (absolute
15    traceable to SI, one standard deviation, or two, or three standard deviations?). The parameter SM is not shown in the map of Figure 1. The accuracy of R1 and R2 is stated to be 10 %. Precipitation is not measured in percentage. What is the unit that has this relative uncertainty? I also think that the table will be more clear if the 3rd line would be on the top (title line) and the acronyms on the line below.

The question of defining accuracy is also valid for the brightness temperatures measured by the radiometer (page 6, line 8).
20    Is 0.5 K an absolute accuracy or x standard deviations?

Author's response:

Table 1 has been changed according to the reviewer's suggestions. The accuracy is a manufacturer information and can't be specified more in detail. The same is true for the given accuracy of the brightness temperature of the WVR.

Comment from Referee:

On page 9 you refer to good agreement twice (lines 15 and 24). The word good is a rather subjective statement and have different meanings to different persons. I think it shall be avoided and instead specify the quality of the agreement in numbers, such as RMS differences and correlation coefficients.

Author's response:

The sentence has been modified accordingly. The good correlation between radiosonde and model data is underlined by explicitly mentioning the mean correlation coefficient.

Comment from Referee:

You comment on that radiometer data are more or less useless to infer water vapour content, liquid water content, and wet delay during rain. Actually, also when large drops of liquid water are present in the sensed volume of air, the accuracy will be worse. A similar effect is if water drops are present on the feed/mirror of the radiometer, which will often be the case for some time also after that the rain has stopped. In Figure 9 you have two outliers (blue dots). It may be worthwhile to investigate if these are in connection to a rain shower or large amounts of the liquid water content.

Author's response:

The 2 outliers in fig. 9 are caused by a previous rain event. The text and fig. 9 were adopted accordingly.

Comment from Referee:

A relevant question for this type of (data) manuscript is how far it is reasonable to take the data analysis? Had it been a regular scientific paper I would have argued that instead of just using retrieval coefficients for the radiometer data from radiosonde data obtained in Munich, it would be required to at least also compare these coefficients from the ones that can be obtained from the launches carried out at the Wettzell site. On the other hand, one reason for publishing a dataset is to inspire others to use it. This could be one such task.

Author's response:

Regarding the use of retrieval coefficients from Munich, we actually intended to derive own retrieval coefficients being valid for the Wettzell site. However, the WVR manufacterer stated that a reliable determination of retrieval coefficients requires continuous radiosonde data over at least 1 year, which were not available at our site. So we chose to use a coefficient set from a neighboring site.

Comment from Referee:

On page 5, line 20 and in Table 5 you use the parameter $Tk_{BB}$ referred to as blackbody temperature which is not defined. Given that it in the dataset is about 10 K warmer than the ambient temperature it cannot be the effective temperature of the atmosphere that is used to calculate the optical depth at the observed frequencies (which also is frequency dependent)?

Author's response:

The parameter $Tk\_BB$ (blackbody temperature) is now explained in the text.

Comment from Referee:

The dataset (described in Table 5) should, where possible, have an uncertainty attached to each parameter. For example, uncertainties in the observed sky brightness temperatures propagate and give, together with uncertainties in the retrieval coefficients, uncertainties in the inferred parameters.

5 Author's response:

Regarding the uncertainties of the inferred WVR-parameters, the following sentence has been added on page 14: "Thus the total accuracy of the estimated water vapour and liquid water content, where uncertainties from the brightness temperature measurement and retrieval coefficients sum up, can't be specified."

Technical corrections:

I find that the font size in all figures is unnecessarily small. The size could in general be say 30-50 % larger in order to improve the readability.

15 Author's response:

The fonts appear larger in the postscript files being submitted with the final version.

All corrections mentioned by the referee were applied.

**Answers to comments of Referee #2**

Comments from Referee:

CONT-17 is the most recent continuous VLBI campaign over two weeks organized by the International VLBI Service for
25 Geodesy and Astrometry (IVS) to assess and push the frontiers of current geodetic VLBI capabilities. For example, it is the ideal test bed to determine high-resolution Earth rotation parameters and other geodetic quantities from three different networks (A, B, and VGOS). One very important error source in VLBI is the modelling of tropospheric delays. Consequently, CONT-17 is perfectly suited to assess the modelled and estimated tropospheric delays at the participating sites, e.g. by comparison with other techniques like GNSS, water vapor radiometers or numerical weather models. In the
30 past, it has always been rather diffcult and cumbersome to collect information from other sources. Here, the authors provide a unique data set to the scientifc community, which can be used for many studies related to the geodetic observatory in Wettzell and CONT-17 in general. In the following, I am going to highlight a few of those: The data set, in particular the radiosonde data but also the weather modes, are well suited to derive the best possible models like mapping functions. These mapping functions can then be used to validate existing models like the Vienna Mapping Functions. Moreover, locally

measured meteorological data are very useful for the determination of local atmospheric ties. The combination of the various data sets can be used to derive information about turbulence, etc.

The manuscript is very clear and well written. I randomly checked the provided datasets on Pangaea, and I could well assess the content. I very much appreciate the possibility to see the data in html and to plot time series. Thanks to the authors and the team at the Geodetic Observatory Wettzell for providing this special and unique dataset! The scientifc community will certainly use the data.

I just found two typos on page 16: contrubution, confindence

Author's response:
The authors are grateful for the comments of the referee.

Author's changes in manuscript:
All corrections mentioned by the referee were applied.

[revised manuscript text omitted]